# Effects of Aroma Foot Massage on Sleep Quality and Constipation Relief among the Older Adults Living in Residential Nursing Facilities

**DOI:** 10.3390/ijerph19095567

**Published:** 2022-05-04

**Authors:** Jung-In Kang, Eun-Hye Lee, Hyeon-Young Kim

**Affiliations:** 1Department of Dermatology, Sahmyook Medical Center, Seoul 02500, Korea; iny519@naver.com; 2College of Nursing, Sahmyook University, Seoul 01795, Korea; leeeh@syu.ac.kr; 3VR Healthcare Content Lab, Sahmyook University, Seoul 01795, Korea

**Keywords:** aroma foot massage, sleep quality, constipation, aged

## Abstract

This study focused on the effects of aroma foot massage on sleep quality and constipation relief among older adult residents in nursing facilities. This research used a non-equivalent control group and a quasi-experimental design. The participants included 40 older adults aged ≥70 years residing in two nursing facilities in Seoul City. The aroma foot massage nursing intervention consisted of a preparation stage using jojoba carrier (aroma recipe) oil and lavender oil, an aroma foot massage stage, and a finishing stage. Sleep quality scores after the experiment increased by 3.72 at post-test (M = 38.44) compared to pre-test (M = 34.72), which confirmed that sleep quality improved significantly following intervention in the experimental group as compared to the control group (F = 14.45, *p* = 0.001). Furthermore, the frequency of defecation in the experimental group was significantly higher than that in the control group (Z = −3.93, *p* < 0.001). Similarly, the constipation assessment scores decreased at post-test significantly by 2.39 in the experimental group as compared to the control group (F = 17.87, *p* < 0.001). These results confirm that aroma foot massage is an effective nursing intervention for alleviating constipation symptoms and improving sleep quality. Therefore, we recommend that aroma foot massage be used as a complementary intervention in combination with drug-based treatment to improve sleep quality and relieve the constipation symptoms experienced by older adults living in nursing facilities.

## 1. Introduction

Given the high prevalence of chronic diseases and dementia in this population, older adults must often rely on others to perform daily activities. As the global population of older individuals increases, so does the number of elderly residents in care facilities [1]. Older adults admitted to residential care facilities experience frequently physical, social, and psychological challenges and changes [2], resulting in various geriatric health problems. In particular, the elderly in residential facilities who perform limited daily physical activity and are highly dependent on others to assist them with physical functions often spend a lot of time in bed. The physical activities of the elderly are limited owing to their increased sedentary lifestyle, and it has been reported that more constipation occurs among older adults in nursing homes. Reduced physical activity and increased time in bed may have negative effects on sleep quality and bowel movements [3,4]. If the quality of sleep and bowel movements are not properly managed, the risks of decreased life satisfaction and reduced quality of life increase [5]. Therefore, constipation and sleep are one of the main problems for the elderly in long-term care facilities.

Age-related changes in sleep patterns are likely to lead to sleep disorders. With increasing age, sleep efficiency relative to sleep time decreases. Insufficient sleep has also been shown to affect the autonomic nervous system, thereby causing constipation or exacerbating existing constipation symptoms [6].

Short-term drug interventions for health problems such as sleep disorders and constipation in older people are effective. However, prolonged drug use can lead to drug resistance, drug misuse and abuse, drug dependence, and impaired cognitive and psychomotor functions. Therefore, non-drug treatment options are considered important [7]. For older adults in residential care, problems related to sleep and constipation are major threats to their physical and psychological health. Therefore, effective and safe alternative therapies without side effects should be considered to enhance the overall quality of life of the elderly.

Among non-drug interventions, aroma massage is known to have greater mental and physical relaxation effects than other massages. The effects of aroma massage may also be maximized through massage contact [8]. Numerous reflex points in the human body correspond to different organs and systems. The feet are known to be the most sensitive body part to physical stimulation. Additionally, there are several reflex points in the feet that correspond to the tissues, organs, and glands of the body. Therefore, stimulating the foot reflex points corresponding to the region from the large intestine to the anus can smooth and regulate bowel function and strengthen the bowel to properly excrete bodily waste [9]. Intensive foot massage at reflex points corresponding to the stomach, large intestine, and anus may effectively relieve constipation [10].

Numerous studies on older adults have found that massage improves sleep quality [11]. However, few studies have investigated the effects of aroma foot massage on sleep and constipation in older adults in residential care. There have been studies on the effects of abdominal or meridian massage on constipation relief in the elderly population, but interventional studies on the effects of foot massage on constipation relief are lacking [12,13]. Few studies have investigated the effects of foot massage on constipation relief, but they have shown inconsistent results, and there have been participant differences in those studies [14,15,16,17]. Through a review of previous studies, it was confirmed that studies that integrated sleep and constipation for the well-being of the elderly in long-term care facilities were limited. We believe that an integrative approach to sleep and constipation will be an important nursing intervention for quality of life in the elderly.

Therefore, we considered it necessary to research the effects of massage on sleep and constipation using an integrated approach. Thus, this study investigated the effects of foot massage using aromatic oils, which is known to be superior to ordinary massage, on sleep quality and constipation among older adult residents of nursing facilities. This interventional study used aroma foot massage as an alternative and complementary therapy to improve the quality of life of older adults residing in care facilities. The hypotheses of this study were as follows:

**Hypothesis** **1.**
*The sleep score of the experimental group using aroma foot massage will be higher than that of the control group not using aroma foot massage.*


**Hypothesis** **2.**
*The number of bowel movements in the experimental group using aroma foot massage will be higher than in the control group not using aroma foot massage.*


**Hypothesis** **3.**
*The constipation assessment score will decrease in the experimental group using aroma foot massage compared to the control group not using aroma foot massage.*


## 2. Materials and Methods

### 2.1. Study Design

This quasi-experimental study used a non-equal control group and a non-synchronized design. This study investigated the effects of aroma foot massage on sleep quality and constipation among older adult residents of care facilities. This study was approved by the Institutional Review Board (IRB) at S University in Seoul, South Korea (IRB approval number: 2-7001793-AB-N-012019068HR). The study period was 16 September 2019, to 11 October 2019. The study procedure was conducted in the order of researchers’ preparation as follows: aroma foot massage, training of research assistants, pre-test, experimental treatment, and post-test. This study followed the procedural guidelines for the design, implementation and reporting of findings of the Consolidated Standards of Reporting Trials (CONSORT) [18].

### 2.2. Participants

The study participants included older adults aged ≥70 years residing in two residential care facilities in Seoul, South Korea, who understood the purpose of the study and consented to participate. The study participants were unable to move and with limited physical activity, provided with an equal meal, and maintained constant management.

The specific selection criteria are as follows:(1)Male and female older adults aged ≥70 years who understood the purpose of the study and agreed to participate.(2)Those with a Korean version of the Mini-Mental State Examination (MMSE-K) score ≥24 or whose cognitive function score was within the normal range.(3)Those who could comprehend the questionnaire content and communicate normally.(4)No olfactory dysfunction or foot lesions.(5)Those who had no symptoms of hypersensitivity reaction 30 min after aroma oil was applied to the inside of their upper arm.(6)Those who had not received any treatment or intervention for their sleep and bowel movements within the last three months.(7)Those with less than two bowel movements per week or a score of four points or more on the constipation assessment scale.

The total number of participants required for this study was calculated with a repeated measure of analysis of variance (ANOVA) using the G*Power 3.1.7 program based on a power (1-β) of 0.95, a significance level (α) of 0.05, an effect size of 0.40, two groups, two repeated measures, and with considerations of within-between interaction. The effect size for sleep and constipation was determined to be 0.40 because this was 1.40–1.64 in a previous study [11] that investigated effects of foot massage on home-dwelling older adults, and which was judged to be large. The calculated number of participants in each group was 16. Considering a 20% dropout rate, 40 participants (20 participants per group) were recruited. Two participants withdrew from the experimental group for personal and other reasons. The final number of participants was 18 and 20 in the experimental and control groups, respectively. To limit the experiment, and for the convenience of providing the intervention and collecting data, those residing in a facility located in D District, Seoul were assigned to the experimental group, while those residing in a facility located in N District, Seoul were assigned to the control group.

### 2.3. Instruments

#### Sleep

Sleep status was measured using the Korean version of the Verran and Snyder-Halpern Sleep Scale developed by Snyder-Halpern and Verran (1987) [19] and modified and supplemented by Oh et al. [20]. This scale is used to measure an individual’s sleep status in the past week and consists of 15 items. Regarding scoring, 4 points were given for “not at all”, 3 for “occasionally”, 2 for “often”, and 1 for “always”. The negative items were inversely converted. The scores ranged from 15 to 60 points. A higher score indicates a higher quality of sleep. The scale is in the form of a self-report questionnaire. Reliability of the scale was confirmed by a Cronbach’s α of 0.79 at the time of the development of the original scale, and a Cronbach’s ⍺ of 0.75 in a study by Oh et al. [20]. Cronbach’s α for this study was 0.82.

### 2.4. Constipation

#### 2.4.1. Number of Bowel Movements

In the context of this study, the number of bowel movements refers to the number of bowel movements of each participant for the past two weeks, as reported by resident caregivers. This was calculated by checking the number of bowel movements of each participant daily and summing the numbers.

#### 2.4.2. Constipation Assessment Score

Constipation was measured using the Korean version of the Constipation Assessment Scale (CAS), which was originally developed by Mcmillan and Williams (1989) [21] and was translated into Korean by Yang et al. [22]. The CAS consists of eight items related to abdominal distention associated with constipation, amount of gas passed rectally, number of bowel movements, stool pattern, discomfort when passing stools, rectal fullness or pressure, amount of stool passed, and ease of passing stool. CAS was used to measure constipation in the previous week. Each item is rated on a 3-point Likert-type scale ranging from 0 (not at all)” to 1 (slightly agree) and 2 (strongly agree)”. The total score ranges from 0 to 16 points, and a score ≥4 points, as measured by this tool, indicates constipation; a higher score indicates more serious constipation. The reliability of this tool was confirmed by a Cronbach’s α of 0.70 at the time of the development of this tool, and a Cronbach’s ⍺ of 0.77 during this study.

#### 2.4.3. Training of Research Assistants

For this study, the researcher completed a training course accredited by the Foot Massage Association and trained two research assistants (foot massage therapists) who held valid foot massage licenses and had more than three years of foot massage experience. The purpose of the training was to familiarize the research assistants (foot massage therapists) with the treatment materials and experimental protocols to ensure homogeneity and increase the reliability of the experiment. The training materials were modified and supplemented based on a literature review of foot care [10] and the practical skills learned by the researcher from foot massage experts. The validity of the content was verified in consultation with three field experts, each of whom had more than 10 years of foot massage experience and held a valid international foot massage license. During the first training session, the purpose and content of this study were explained to the research assistants. During the second training session, the treatment tools, methods, and time were explained, and the research assistants were instructed to perform a preliminary/practice treatment. Subsequently, problems identified during preliminary treatment were assessed and corrected before conducting the actual treatment.

#### 2.4.4. Pre-Test

To recruit the study participants, nursing homes were conveniently sampled from two institutions with as similar conditions as possible, and the researcher randomly selected two residential facilities in Seoul, which were similar in their usage patterns and socio-economic conditions, thus ensuring homogeneity between the two facilities. The researcher visited both the selected facilities to explain to their respective heads the purpose and procedure of the study before seeking their cooperation to conduct the research. After receiving permission to perform the experimental intervention, participants were recruited using convenience sampling for ease of data collection. To reduce the variables, we excluded subjects who received sleep-or bowel-related treatment or other interventional therapy within three months of the study period. Recruitment notices were displayed at both facilities. The purpose and methods of the study were explained to the residents: those who understood the study and were interested in participating voluntarily. A written consent form describing the purpose and procedure of this study was provided to the participants. The participants in both the experimental and control groups received detailed information about the purpose, method, procedures, and expected effects of this study, and they provided consent via the nurses in charge at their respective facilities. A completed questionnaire regarding their general characteristics, sleep, and constipation assessment scores was obtained. Furthermore, the questionnaire for the study participants calculated the pre-test sleep score using a sleep diary. Data on the number of bowel movements were calculated based on the bowel diary completed by the nurses in charge at the two facilities.

#### 2.4.5. Experimental Treatment

This study investigated the effects of foot massage using lavender aroma oil (Lavandula angustifolia), whose positive effects have been demonstrated by several previous studies [11,13,17,23]. Foot massage with aromatic oils is suitable for both children and older adults. In this study, oil diluted with 200 mL of jojoba oil (*Simmondsia chinensis*) in 2 mL of 100% lavender undiluted solution was used [24]. This involved a therapist smoothly stroking or pressing and rubbing the reflex points and muscles in the feet using their hands and without special tools to avoid side effects [10]. Considering that this study involved older adults residing in care facilities, and based on advice from foot massage experts and findings of previous studies, the solution was applied eight times for a total of four weeks, and each session was conducted for 30 min. A general aroma foot massage intervention was used in this study by picking the pulse, which massaged the legs and feet as a whole, and finishing with a raisin massage, stimulating the foot reflexes for each part.

#### 2.4.6. Post-Test

The researcher checked the nurses in charge of the experimental and control groups regarding the number of bowel movements per week for two weeks after the end of the experimental treatment. With the help of the nurses in charge, who provided the data on sleep and constipation assessment scores before the experiment, sleep scores at post-test were calculated through post-test questions given to participants. The researcher recorded the sleep and constipation assessment scores for those in the experimental group using the same questionnaire. The measurements were conducted twice, during the pre-and post-examination periods. The control group was investigated in the same manner.

### 2.5. Data Analysis

The collected data were analyzed using SPSS Statistics version 25.0. The general characteristics of the participants and dependent variables were presented as numbers, percentages, means, and standard deviations. Pre-homogeneity between the two groups was measured using the chi-squared test, Fisher’s exact test, and independent *t*-test. Normality tests were performed for dependent variables, such as sleep score, number of bowel movements, and constipation assessment score, using the Shapiro–Wilk test. The differences between the variables before and after aroma foot massage in the experimental and control groups were analyzed using repeated-measures ANOVA and the Mann–Whitney U test. The reliability of each tool used in this study was verified using Cronbach’s α. *p* value of less than 0.05 was considered significant.

## 3. Results

### 3.1. Test for the Homogeneity of General Characteristics between the Experimental and Control Groups

A total of 38 older adults (18 in the experimental group and 20 in the control group) participated in the post-test after the experiment ended. These included five (13.2%) male and 33 (86.8%) female elderly participants. There were five (13.2%) participants aged 70–79 years, 24 (63.2%) participants aged 80–89 years; and nine (23.72%) participants aged ≥90 years. In terms of education, eight (21.1%) had no formal education, 14 (36.8%) were primary school graduates, 11 (28.9%) were middle school graduates, and five (13.2%) were high school graduates or higher. The duration of residence at the care facility was ≤1 year for nine (23.7%) participants, 1–2 years for eight (21.1%) participants, 3–4 years for 15 (39.5%) participants, and ≥8 years for six (15.8%) participants. With regard to the number of family visits, nine (23.7%) participants had a family visit once a week, eight (21.1%) had once every two weeks, one (2.6%) had once every three weeks, five (13.2%) had once every four weeks, and 15 (39.5%) had other numbers of family visits. In terms of subjective health status, nine (23.7%) participants reported that their subjective health status was “good;” 28 (73.7%) reported that it was “normal;” and one (2.6%) reported that it was “bad.” In terms of disease diagnosis, 27 (71.1%) participants had cerebrovascular disease; three (7.9%) had heart disease; eight (21.1%) had other diseases. There were no statistically significant differences in any of the variables when the general characteristics of the participants in the experimental and control groups were compared, indicating that the two groups were homogeneous (Table 1).

### 3.2. Test for the Homogeneity of Dependent Variables between the Experimental and Control Groups

There was no significant difference in the sleep scores among the participants before the experiment (34.72 points in the experimental group and 37.20 points in the control group), confirming that the two groups were homogeneous (t = −1.37, *p* = 0.178). The mean number of pre-test bowel movements was 5.56 in the experimental group and 5.00 in the control group, respectively, indicating that the two groups were homogeneous (t = 1.34, *p* = 0.188). Furthermore, there was no significant difference in constipation assessment scores between the two groups (8.28 points in the experimental group and 8.25 points in the control group), indicating that the two groups were homogeneous (t = 0.49, *p* = 0.961) (Table 2).

### 3.3. Hypothesis Testing

**Hypothesis** **1.***The sleep score of the experimental group using aroma foot massage will be higher than that of the control group not using aroma foot massage*.

Changes in sleep scores were detected after the experiment, which suggests significant interactions between the groups and time points (pre-test and post-test). The results indicated that the changes between the experimental and control groups differed according to the time points (F = 14.45, *p* = 0.001). After adjusting for the interactions between the groups and time points, the sleep score in the experimental group increased by 3.72 points from the pre-test (M = 38.44) to the post-test (M = 34.72) (*p* = 0.003). In contrast, the sleep score in the control group decreased by 0.80 points from pre-test (M = 37.20) to post-test (M = 36.40) (*p* = 0.186). The sleep score in the experimental group increased substantially compared with that in the control group, thereby supporting Hypothesis 1 (Table 3).

**Hypothesis** **2.**
*The number of bowel movements in the experimental group using aroma foot massage will be higher than in the control group not using aroma foot massage.*


As the number of bowel movement variables did not satisfy a normal distribution, non-parametric analyses were used. The number of bowel movements in Table 4 was measured over two weeks. The number of bowel movements in the experimental group using aroma foot massage increased from 5.56 to 6.94, which was significantly higher when compared to an increase from 5.00 to 5.15 in the control group (Z = −3.93, *p* < 0.001), thereby supporting Hypothesis 2 (Table 4).

**Hypothesis** **3.**
*The constipation assessment score will decrease in the experimental group using aroma foot massage compared to the control group not using aroma foot massage.*


Changes in constipation assessment scores after the experiment revealed significant interactions between the groups and time points. The results confirmed that the changes in the experimental and control groups differed according to the time-point (F = 17.87, *p* < 0.001). After adjusting for interactions between the groups and time points, the results showed that the constipation assessment score in the experimental group decreased by 2.39 points (*p* < 0.001) in the post-test (M = 5.89) compared to the pre-test (M = 8.28) (*p* < 0.001), and the constipation assessment score in the control group decreased by 0.60 in the post-test (M = 7.65) compared to the pre-test (M = 8.25). In other words, the constipation assessment score in the experimental group decreased substantially compared with that in the control group, thereby supporting Hypothesis 3 (Table 5).

## 4. Discussion

This study aimed to investigate the effects of aroma foot massage on sleep quality and constipation in older adults living in residential care facilities. The findings of this study are as follows.

### 4.1. Effects of Aroma Foot Massage on Sleep Quality

The results of testing Hypothesis 1 revealed that the sleep score in the experimental group using aroma foot massage increased during the post-test (M = 38.44) compared to the pre-test (M = 34.72). The results demonstrated a significant difference in sleep satisfaction between the groups (F = 14.45, *p* = 0.001), which concurred with previous studies that concluded that aroma foot massage improved sleep quality [11].

In contrast to the results of this study, Song [25] reported that the effects of ordinary massage without aroma oil on sleep in home-dwelling older adults were not significant, and sleep quality was not improved. Lavender oil, which has proven calming, sedative, and physical relaxation effects [26], was used in this study. The results of the present study demonstrate that foot massage using aroma oil is more effective than ordinary massage. This finding was consistent with the findings by Seo et al. [27], which showed that when a blend of lavender, bergamot, and chamomile oils (2:2:2 blending ratio) was used for aroma hand massage for elderly residents of welfare facilities three times a week for two weeks, their sleep status was significantly enhanced. Seo et al.’s results were also consistent with the results of Park et al. [28] who reported that, when a blend of lavender and bergamot oils (1:1 blending ratio) was used for aroma hand massage in hospitalized elderly patients three times a week for two weeks, their sleep quality improved significantly. They further suggested that certain types of oils have significant effects on sleep improvement.

Won [29] used peppermint, eucalyptus, and rosemary oils for foot massages and reported that the sleep score of participants increased slightly from 25.9 to 38.05 points. However, this difference was not statistically significant. The results of this study indicate that lavender oil may be more effective in improving sleep quality. Several studies have reported that linalool, a component of lavender oil used in this study, has soothing and sleep improvement effects [30,31] because it regulates the autonomic nervous system and that lavender is among the most preferred scents among older adults. It has also been proven that different types of lavender oil provide different levels of relaxation effects, indicating that aroma massage has its own effects and not just contact or placebo effects [26].

Foot massage intensively stimulates blood flow in the foot area to increase blood circulation throughout the body [31], while also providing physical and psychological relaxation [8], thereby improving sleep and sleep patterns [32]. A study by Noh [33] reported the effects of aroma hand massage on sleep status in hospitalized elderly patients in a long-term care hospital. The results showed that the sleep score slightly increased from 35.07 points before the experiment to 38.19 points after the experiment; however, the results were not statistically significant. The authors suggested that the relaxation effects of foot massage, together with the sleep improvement effects of lavender oil, contributed to the enhancement of sleep quality in this study.

The study participants who received aroma foot massage described the experience as “it’s cool,” “it’s good,” “I’m getting sleepy,” or “I feel like my husband is touching me.” One participant even farted while receiving a foot massage. During the aroma foot massage sessions, their anxieties and stresses were ameliorated, their serenity was enhanced, and they reported psychological well-being and comfort as a result of the massage therapists touching their feet. The synergistic effects of foot massage and lavender oil improved the sleep quality of participants. One participant in the experimental group had severe edema in both legs. As the participant’s blood vessels were relaxed through aroma foot massage, better blood circulation and increased lymphatic flow were promoted [34], thereby reducing edema.

### 4.2. Effects of Aroma Foot Massage on Constipation

The results of testing Hypothesis 2 revealed that the participants’ number of bowel movements increased from 5.56 to 6.94%, which was statistically significant. This finding was consistent with the results of two studies. The first is a study by Min [16], which showed that reflexology foot massage had constipation-relief effects among dance students. The second is a study by Park [14], which reported that the number of bowel movements increased from 2.29 before foot reflex massage, to 4.48 after one week of foot reflex massage in the case of hospitalized stroke patients under rehabilitation. Massage reduces muscle tension, increases blood and lymph circulation, decreases heart rate and blood pressure, and improves body flexibility [10]. Thus, the aforementioned results suggest that aroma foot massage is effective in increasing the number of bowel movements.

The results of testing Hypothesis 3 revealed that the constipation assessment score significantly decreased in the experimental group as compared to the control group, thereby supporting Hypothesis 3. This finding was consistent with a study by Kim & Kang [15] which reported on the effects of foot reflex massage on sleep, depression, and constipation in community-dwelling elderly adults. Kim & Kang indicated that foot reflex massage significantly affected sleep, depression, and constipation.

Although several studies have been conducted regarding the effects of foot massage on constipation in home-dwelling elderly adults [15], female college students [17], and stroke patients [14], there have been no studies on the effects of foot massage on constipation in older adults residing in care facilities, to the best of our knowledge. The results of this study involving older adults residing in care facilities found that aroma foot massage relieved constipation, which is consistent with the results of previous studies involving different participants.

These results may be explained by the effects of aroma foot massage on peripheral nerves, which improve blood circulation and help eliminate waste or precipitants [10].

Foot reflexology is considered effective in relieving constipation because massaging reflex points in the feet improves blood circulation, facilitates waste elimination, relieves anxiety and tension, and provides psychological relaxation and a sense of well-being [35]. In this study, the effects of aroma foot massage were maximized in the experimental group by increasing the focused stimulation of the reflex points in the foot area that corresponded to the digestive system.

Previous studies on the use of interventions to relieve constipation included a study that examined the effects of general abdominal massage without aroma [36] and a study by Kim [37], which showed the effects of abdominal meridian massage in relieving constipation in hemiplegic patients. However, abdominal massage has the risk of lowering body temperature because of excessive exposure and is limited by the areas that can be massaged. In contrast, foot massage can be used more effectively and widely, thereby optimizing the cost-effectiveness of treatment.

This study used an integrated approach to investigate the use of foot massage as an intervention for improving sleep and alleviating constipation in elderly residents of care facilities, and demonstrated its effects on sleep and constipation. Future studies are required to supplement and develop nursing interventions and procedures to make them more effective and accessible.

This study only assessed the subjective sleep quality of the participants. The study was limited in that it did not measure the physiological changes or specific effect mechanisms of the sleep patterns. Although foot massage using lavender oil has been proven effective in improving the sleep quality of older adults residing in care facilities, future studies using different types of aroma oils, including those reported in past studies, are required to compare the differences in sleep improvement when providing aroma foot massage using various types of aroma oils. Furthermore, future studies involving elderly care facility residents and the examination of physiological variables or other measurement methods are warranted. With more research evidence, the specific mechanisms underlying the effects of various types of aroma oils can be elucidated and compared.

### 4.3. Limitation

This study has limitations in that there was no placebo intervention in the control group during the duration of the study and blinding in the intervention group was not maintained. Furthermore, there was a difference between the facilities where the intervention was made because the third variable was not completely controlled due to various life-related events that could occur. There are not many studies that have verified the effects of aroma foot massage on sleep and constipation in the elderly living in facilities at the same time. It is necessary to conduct repeated research to confirm the validity of this study by expanding the research subject and scope, and strengthening the research design. Since this study was conducted only for the elderly living in facilities at two institutions, it is necessary to apply it to the elderly population in other facilities. Although it is a study targeting the elderly aged 70 and over, due to the characteristics of the elderly living in facilities aged 80 years old or more, there are many older subjects, so a study that subdivides age and gender is necessary.

## 5. Conclusions

The results of this study demonstrated that the sleep pattern score significantly improved in the experimental group using aroma foot massage compared to the control group not using aroma foot massage and that the constipation assessment score decreased in the experimental group compared to the control group. These findings suggest that aroma foot massage is effective in improving sleep quality and alleviating constipation in elderly residents of care facilities.

This study is significant in that it used an integrated approach to investigate the physiological problems of sleep and constipation in older adults through aroma foot massage, and demonstrated its effects and possible mechanisms. The results contribute to nursing practice and other studies on nursing interventions for older adults residing in care facilities.

Aroma foot massage was found to be an effective intervention for improving sleep quality and alleviating constipation in elderly residents of care facilities through its psychological and physiological relaxation and intestinal stimulating effects. It is expected that aroma foot massage may be used as a complementary and alternative therapy in nursing interventions.

This study investigated the effects of an aroma foot massage nursing intervention in older adults residing in two care facilities in Seoul, South Korea. Therefore, the generalizability of these results should be considered with caution. The long-term effects of the aroma foot massage program have not been verified. Therefore, further studies using different research methods and sample populations are required to confirm their effects.

## Figures and Tables

**Table 1 ijerph-19-05567-t001:** Homogeneity Test According to the General Characteristics of the Experimental Group and the Control Group (*n* = 38).

General Characteristics	Assortment	Experimental Group (*n* = 18)	Control Group (*n* = 20)	χ^2^-Test/ Fisher’s Exact	*p*
		(%)	(%)		
Sex ^†^	Male	3 (16.7)	2 (10.0)	0.37	0.653
	Female	15 (83.3)	18 (90.0)		
Age ^†^	70–79	3 (16.7)	2 (10.0)	0.37	0.899
	80–89	11 (61.1)	13 (65.0)		
	≥90 years	4 (22.2)	5 (25.0)		
Education ^†^	Uneducated	3 (16.7)	5 (25.0)	0.97	0.833
	Elementary school	6 (33.3)	8 (40.0)		
	Middle school	6 (33.3)	5 (25.0)		
	High school or higher	3 (16.7)	2 (10.0)		
Residence period ^†^	<1	3 (16.7)	6 (30.0)	2.00	0.629
	1–2 years	3 (16.7)	5 (25.0)		
	3–4 years	9 (50.0)	6 (30.0)		
	>5 years	3 (16.7)	3 (15.0)		
Family visits ^†^	Once a week	5 (27.8)	4 (20.0)	3.38	0.581
	Once every 2 weeks	3 (16.7)	5 (25.0)		
	Once every 3 weeks	1 (5.6)	0 (0.0)		
	Once every 4 weeks	1 (5.6)	4 (20.0)		
	Other	8 (44.4)	7 (35.0)		
Subjective health ^†^	Good	6 (33.3)	3 (15.0)	2.47	0.260
	Normal level	12 (66.7)	16 (80.0)		
	Bad	0 (0.0)	1 (5.0)		
Disease classification ^†^	Cerebrovascular disease	13 (72.2)	14 (70.0)	0.77	0.766
	Heart disease	2 (11.1)	1 (5.0)		
	Other	3 (16.7)	5 (25.0)		

^†^ Fisher’s exact test.

**Table 2 ijerph-19-05567-t002:** Homogeneity Test for the Dependent Variables of the Experimental Group and the Control Group (*n* = 38).

Item	Experimental Group (*n* = 18)	Control Group (*n* = 20)	t	*p*
	M (±SD)	M (±SD)		
Sleep score	34.72 (±5.53)	37.20 (±5.57)	−1.37	0.178
Number of feces	5.56 (±1.42)	5.00 (±1.12)	1.34	0.188
Constipation assessment score	8.28 (±1.45)	8.25 (±2.00)	0.49	0.961

**Table 3 ijerph-19-05567-t003:** Inter-group Difference Test for Sleep Score (*n* = 38).

Item	Assortment (*n*)	Before Experiment M (±SD)	After Experiment M (±SD)	Source	F	*p*
Sleep score	Experimental group (*n* = 18)	*t* = 3.46 *p* = 0.003		Group	0.02	0.003
		34.72(±5.53)	38.44(±2.43)	time-point	6.03	0.186
	Control group (*n* = 20)	*t* = 1.37 *p* = 0.186		Group * time-point	14.45	0.001
		37.20(±5.57)	36.40(±4.99)			

**Table 4 ijerph-19-05567-t004:** Between Groups Differences Test for Stool Count (*n* = 38).

Item	Experimental Group (*n* = 18)	Control Group (*n* = 20)	Z	*p*
	M (±SD)	M (±SD)		
Stool Count ^†^	6.94 (±1.21)	5.15 (±1.09)	−3.93	<0.001

^†^ Mann–Whitney U test.

**Table 5 ijerph-19-05567-t005:** Test for Differences Between Groups for Constipation Assessment Scores (*n* = 38).

Item	Assortment (*n*)	Before Experiment M (±SD)	After Experiment M (±SD)	Source	*F*	*p*
Constipation assessment scores	Experimental group (18)	*t* = –7.14 *p* < 0.001		Group	2.96	0.036
		8.28 (±1.45)	5.89 (±0.90)	time-point	49.88	<0.001
	Control group (20)	*t* = –2.26 *p* = 0.036		Group * time-point	17.87	<0.001
		8.25 (±2.00)	7.65 (±2.03)			

## Data Availability

The data presented in this study are available on request from the corresponding author.

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
