# Peer review of "Effects of Aroma Foot Massage on Sleep Quality and Constipation Relief among the Older Adults Living in Residential Nursing Facilities"

_ijerph, 2022, doi:10.3390/ijerph19095567_

Round 1

Reviewer 1 Report

The authors have made the needed changes in response to reviewer comments.

Author Response

Thank you for your prompt and positive review.

Reviewer 2 Report

Thank you for inviting me to review the new version of the manuscript entitled “Effects of aroma foot massage on sleep quality and constipation relief among the older adults living in residential nursing facilities”

1. It seems that jojoba carrier oil and lavender oil are popular names of natural products obtained from plants. I suggest including the species of plants for each oil.

2. Regarding the following question asked in the first review: “In the introduction section, authors have presented several statements with similar ideas, or words in repetitions. When they reported previous studies, comparing the data, there is a lack of information regarding the difference between these studies. There is a general presentation that can be difficult to interpret.” Authors did not answer properly. They include statements about COVID-19 pandemic. The study was performed before the pandemic. 

3. P-value was not reported in the Data Analysis (methods section).

4. Limitations should be more explored.

5. This study is a clinical trial by using natural products. The Enhancing the QUAlity and Transparency Of Health Research (EQUATOR) presents many guidelines for reporting studies involving human beings, including clinical trials. None of those was reported in this manuscript.

6. In addition, a Clinical Trial Registry was not presented by the authors in this manuscript.

Author Response

Dear Editor and Reviewers,

We wish to thank you for your thoughtful comments and valuable feedback on the manuscript originally titled, “Effects of aroma foot massage on sleep quality and constipation relief among the older adults living in residential nursing facilities”

We have modified the manuscript according to your suggestions, rewriting and rephrasing sections to improve clarity, adding further information, and explaining in detail the points that were previously vague. For your convenience, we have set the revisions in the resubmit manuscript. We believe that the revised version of this paper will be of interest to the readership of the International Journal of Environmental Research and Public Health.

-----------------------------------------------------------------------------------------------------

Reviewer 2’s comment

Comments and Suggestions for Authors

Point 1. It seems that jojoba carrier oil and lavender oil are popular names of natural products obtained from plants. I suggest including the species of plants for each oil.

  • Response: Thank you for your close review point, we’ve added the specific species of plants for each oil.

On page 5:

  • Lines 214-218: This study investigated the effects of foot massage using lavender aroma oil(Lavandula angustifolia), whose positive effects have been demonstrated by several previous studies (11, 13, 17, 23). Foot massage with aromatic oils is suitable for both children and older adults. In this study, oil diluted with 200 ml of jojoba oil(Simmondsia chinensis) in 2 ml of 100% lavender undiluted solution was used(24).

Point 2. Regarding the following question asked in the first review: “In the introduction section, authors have presented several statements with similar ideas, or words in repetitions. When they reported previous studies, comparing the data, there is a lack of information regarding the difference between these studies. There is a general presentation that can be difficult to interpret.” Authors did not answer properly. They include statements about COVID-19 pandemic. The study was performed before the pandemic.  

  • Response: We further described the differences from previous studies and the significance of the study.

On page 1:

  • Lines 37-43: The physical activities of the elderly were limited owing to the increased sedentary lifestyle, and it has been reported that more constipation occurs among the older adults in nursing homes. Reduced physical activity and increased time in bed may have negative effects on sleep quality and bowel movements (3, 4). If the quality of sleep and bowel movements are not properly managed, the risks of decreased life satisfaction and reduced quality of life increase (5). Therefore, constipation and sleep are one of the main problems for the elderly in long-term care facilities for the elderly.

On page 2:

  • Lines 73-76: Through a review of previous studies, it was confirmed that studies that integrated sleep and constipation for the well-being of the elderly in long-term care facilities were limited. We believe that an integrative approach to sleep and constipation will be an important nursing intervention for quality of life in the elderly.

Point 3. P-value was not reported in the Data Analysis (methods section).

  • Response: Thank you for your detail review point, we added the explanation of p-value.

On page 5:

  • Lines 246-247: P value of less than .05 was considered significant.

Point 4. Limitations should be more explored.

  • Response: We added the limitation for this study through deep insight according to your consideration review.

On page 11:

  • Lines 439-447: There are not many studies that have verified the effects of aroma foot massage on sleep and constipation in the elderly living in facilities at the same time. It is necessary to conduct repeated research to confirm the validity of this study by expanding the research subject and scope, and strengthening the research design. Since this study was conducted only for the elderly living in facilities at two institutions, it is necessary to apply it to the elderly population in other facilities. Although it is a study targeting the elderly aged 70 and over, due to the characteristics of the el-derly living in facilities, the age of 80 years old There are many older subjects, so a study that subdivides the width of age and gender is necessary.

Point 5.  This study is a clinical trial by using natural products. The Enhancing the QUAlity and Transparency Of Health Research (EQUATOR) presents many guidelines for reporting studies involving human beings, including clinical trials. None of those was reported in this manuscript.

  • Response: Thank you for taking a close look. We added the procedure of our study.

On page 3:

  • Lines 101-104: This study was followed the procedural guidelines for the design, implementation and reporting of findings of the Consolidated Standards of Reporting Trials(CONSORT)(18).

Point 6.  In addition, a Clinical Trial Registry was not presented by the authors in this manuscript.

  • Response: Thank you for your reviewing, there is no change in the manuscripts, and please refer to our answer.

Answer:

  • We strongly pursued ethical considerations regarding the conduct of research. However, we did not proceed with the clinical registry in advance, and it is not intended for selective reporting. We will carefully consider your guidelines and reflect on them in future research.

Thank you very much for your prompt and positive review.

This manuscript is a resubmission of an earlier submission. The following is a list of the peer review reports and author responses from that submission.

Round 1

Reviewer 1 Report

Thank you for the opportunity to review this quasi-experimental study on N=40 elderly patients titled, “Effects of aroma foot massage on sleep quality and constipation relief among the elderly living in residential nursing facilities.” Please find my comments below:

Major

  • Please provide more detail related to how number of bowel movements was determined. Was this from the medical record at the facility? Was there any other data such as size and consistency that may aid in determining constipation relief?
  • The authors state they chose 2 residential facilities at random, but they just happen to be similar in usage patterns and socio-economic conditions? Please expand on this and describe how they were “randomly” determined.
  • Please provide more information related to the foot massage. What was the sequence, for how long and how often? Was the exact same procedure followed each time for each participant? What was the brand of the oil?
  • At what frequency was the sleep and constipation assessments made, there is lack of detail in the study methods related to when the pre-intervention and post-intervention assessments were made. It was unclear if post-questionnaires were given? There is only information related to number of bowel movements.
  • Overall, more clarity and detail in the methods section is needed to clearly understand the study plan and execution.
  • In the results section, is table 4 daily or weekly bowel movements?
  • Please add a limitations section. There are several notable limitations that should be included. Namely the lack of blinding in the intervention group, there was no sham intervention for the control group, and differences between facilities.

Minor

  • Line 197-198: Please add citation, and change “proven” to some other descriptor such as consistent with, or demonstrated, etc. Proven is a strong word, the literature is not consistent with that description.
  • I know patients were excluded if they received meds for sleep/constipation, but it would be helpful to add a sentence related to if any participants received an intervention during the study period.
  • Can you provide any information on other activities such as exercise, bathing, etc. that may influence sleep and BMs? Even a sentence that usual care was maintained throughout the study period would be helpful.

Reviewer 2 Report

Thank you for inviting me to review the manuscript entitled “Effects of aroma foot massage on sleep quality and constipation relief among the elderly living in residential nursing facilities” (ID ijerph-1196731) submitted by Kang and Kim. The authors have demonstrated the potential effect of natural medicine for the amelioration of sleep, reduction of constipation and bowel movements improvement in elderly, using lavender essential oil.  

Regarding the abstract section, details for the methods and statistical analysis should be provided by the authors.

In the introduction section, authors have presented several statements with similar ideas, or words in repetitions. When they reported previous studies, comparing the data, there is a lack of information regarding the difference between these studies. There is a general presentation that can be difficult to interpret.  

Regarding the instrument for sleeping status, by Verran and Snyder (1987), the reference was not reported.

The main weakness is in the methods. Dietary habits are an important factor for regulation of bowel movements and constipation (https://doi.org/10.1159/000501961). The authors have not considered this variable in their study. In this way, there is a concern about the food intake by elderly in the experiment days. 

The authors reported that lavender oil has been used in several studies, but they did not report them (lines 197-198).

In the data analysis section, there is a description for general characteristics. In addition, the first painel of results are concentrated in these sociodemographic data. However, there is a lack of information regarding the type of characteristics in the methods sections. 

In line 233, authors reported 632%. They could clarify that finding. 

It seems that the number of bowel movements was not higher as authors have reported (6.94 vs. 5.56). There is a statistical difference between them. However, it seems that essential oil therapy improved only 1.2 times as compared to before the experiment.

References must be verified and corrected. In the discussion section, authors described a study by Kim and Kang, number 14, in contrast with the following study reported in the references section:

  1. Kuroda, K.; Inoue, N.; Ito, Y.; Kubota, K.; Sugimoto, A.; Kakuda, T.; et al. Sedative Effects of the Jasmine Yea Odor and (R)-472 Linalool, One of its Major Odor Components, on Autonomic Nerve Activity and Mood States. European Journal of Applied 473 Physiology 2005, 95, 107-114.

According to references section:

Line 318, “...This finding was consistent with the findings of a study by Suh et al. [20]”, the following reference number 20 is:

  1. No, S.Y. The Effect of Aroma Massage on the Itching, Skin pH and Skin Moisture Retentin and Sleep of the Elderly in the 485 Nursing Hospital [dissertation]. Gwangju: Chosun University 2012.

Line 391, “... study by Kim [32], the following reference number 32 is:

  1. Yang, S.; Yoo, S.J. Effects of Dietary Supplements on Constipation After Taking Antidepressants in Depressed Patients. 507 Journal of Psychiatric Nursing 1996, 5, 13-26.

Lines 359-362, “...The first is a study by Min [28], which showed that reflexology foot massage had constipation-relief effects among dance students. The second is a study by Park [15], the following references numbers 15 and 28 are:

  1. Vlachantoni, A.; Shaw, R.; Willis, R.; Evandrou, M.; Falkingham, J.; Luff, R. Measuring Unmet Need for Social Care 475 Amongst Older People. Population Trends 2011, 145, 56-72.
  2. Vickers, A. Massage and Aromatherapy: A Guide for Health Professional. Chapman and Hall 1996.

In sum, the contrast presented in the references is also a weakness in this study. Authors must verify carefully if those reports match with the ideas written in the manuscript. 

Also, I suggest editing help from someone with full professional proficiency in English.